# Filter Media-Packed Bed Reactor Fortification with Biochar to Enhance Wastewater Quality

**Ezekiel Kholoma [1], Agnieszka Renman [2],*** and **Gunno Renman [2]**

[1] Department of Civil & Environmental Engineering, Botswana International University of Science & Technology, Private Bag 16, Palapye, Botswana; kholomae@biust.ac.bw

[2] Division of Water and Environmental Engineering, KTH Royal Institute of Technology, 100 44 Stockholm, Sweden; gunno@kth.se

* Correspondence: agak@kth.se; Tel.: +46-87906768



**Featured Application: The described filter technology is intended for common use in small-scale wastewater treatment after full-scale system development and field tests.**

**Abstract:** Contamination of water sources by inappropriately disposed poorly treated wastewater from countryside establishments is a worldwide challenge. This study tested the effectiveness of retrofitting sand (Sa)- and gas–concrete (GC)-packed reactors with biochar (C) in removing turbidity, dissolved organic carbon (DOC), phosphate ($PO_4^{3-}$), and total phosphorus ($P_{tot}$) from wastewater. The down-flow reactors were each intermittently loaded with 0.063 L/d for 399 days. In general, all reactors achieved <3 NTU (Nephelometric Turbidity Units) effluent turbidity (99% efficiency). GC reactors dominated in inlet $PO_4^{3-}$ (6.1 mg/L) and DOC (25.3 mg/L) reduction, trapping >95% and >60%, respectively. Compared to Sa ($PO_4^{3-}$: 35%, DOC: 52%), the fortified sand (SaC) filter attenuated more $PO_4^{3-}$ (>42%) and DOC (>58%). Student t-tests revealed that C significantly improved the Sa $PO_4^{3-}$ ($p = 0.022$) and DOC ($p = 0.034$) removal efficacy. From regression analysis, 53%, 81%, and 85% $PO_4^{3-}$ sorption variation in Sa, C, and SaC, respectively, were explained by variation in their effluent pH measures. Similarly, a strong linear correlation occurred between $PO_4^{3-}$ sorption efficiency and pH of fortified ($r > 0.7$) and reference ($r = 0.6$) GC filters thus suggesting chemisorption mechanisms. Therefore, whereby only sand may be available for treating septic tank effluents, fortifying it with biochar may be a possible measure to improve its efficacy.

**Keywords:** wastewater treatment; biochar; packed bed reactor; fortification; phosphorus; dissolved organic carbon

## 1. Introduction

The septic system has proved to be feasible for sewage handling in remote and rural areas [1]. However, even though widely used, research has shown that septic systems contribute largely to loading of nutrients [2], pathogens [3], and organic compounds [4] in surface and ground water sources. To protect human and aquatic lives, environmental protection authorities in many countries recently started to enforce stringent federal discharge quality standards in those areas. For example, the allowable disposal limit for effluent phosphorus (P) in Denmark [5], Norway [6], and Sweden [7] is ≤1 mg/L. Meanwhile, alternative effective technologies, with which the treatment facilities used in remote and rural areas (mostly septic systems) could be replaced or improved, are scarce.

Dissolved organic carbon (DOC) and nutrients are common constituents of septic tank effluent (STE) which often escape treatment in soil treatment units (STU) and ultimately end up in nearby surface water bodies. Since these can pass through 0.45 μm pores, common coarse media, such as

inert fluviatile sand and gravel used for constructing STUs and soak-aways, are generally less capable of attenuating much of their concentrations. The nutrients, that is phosphorus (P) and nitrogen (N), nourish aquatic plants and thus fuel algae blooms. This often leads to eutrophication and hypoxia conditions that are detrimental to aquatic biota. In centralized treatment systems, high P removal is mostly accomplished by processes which employ mechanisms that convert soluble reactive P ($PO_4^{3-}$) to particulate form before filtration, and microbial assimilation [8]. DOC is most commonly removed by use of granulated activated carbon (GAC). The drawback of all these technologies is that they are costly (for example purchasing mixing tanks, chemicals, feeding systems, etc.) [9]. They also produce large amounts of sludge requiring special handling. For example, precipitating $PO_4^{3-}$ in 1 m$^3$ of wastewater to calcium hydroxyapatite ($Ca_5(PO_4)_3OH$) using slaked lime ($Ca(OH)_2$) requires about 0.3–0.6 kg of $Ca(OH)_2$. This in turn produces about 4–10 L of sludge [10]. In general, these are unaffordable to most rural communities. A few supposedly feasible innovations are available for improving on-site wastewater treatment (OWT) systems. These include sequential batch reactors (SBR) or package treatment plants (PTP) [11] for substituting septic systems, multi-chamber tanks or anaerobic baffled reactors [12] for replacing the two-chamber septic tanks and hybrid constructed wetlands for post-secondary treatment. Being that advanced requires expert knowledge for their operation and routine maintenance [13]. Therefore, they may be unsuitable for some would-be users, especially single households. However, as efforts to acquire more efficient technologies continue to be made, an increasing interest seems to be shown in the use of so-called reactive filter (RF) media as retrofits for septic systems. The most common of these materials are the granulated mineral-based media with ionic surface composites, which can react with and chemically or physically bind some wastewater constituents [14]. For example, those used for P removal comprise ionic compounds of calcium (Ca), iron (Fe), aluminum (Al), or magnesium (Mg). Soluble reactive phosphorus ($PO_4'$) has a high affinity for these cations [15]. In principle, as wastewater infiltrates a granulated RF, the targeted wastewater constituent may react with or be exchanged with exposed ligands on surfaces of its particles and thus become part of either surface complexes or precipitates formed. Another important property of RF media responsible for their high sorption capacity is high porosity. This contributes to both high pore volume for holding more water and physical straining of larger diameter particles in wastewater.

Numerous RF media were tested in field-scale studies for their capability of removing nutrients from wastewater [14]. Examples are lava sand [16], Polonite® [17], shell sand [18], Filtralite® [13], and slags [19]. Since bad color and odor are common in STEs, using RF media that are versatile at removing not only P but also DOC may prove to be promising. Biochar, a biomass-derived material, has been found to be versatile in various applications. Examples include carbon dioxide sequestration and reduction of greenhouse gas emission in climate change mitigation [20], soil amelioration, and contaminant immobilization in agriculture [21]. Learning from this, a number of studies tested its capacity in removing nutrients [22], metals [23], organic compounds [24], and pathogens [25] from wastewater. Promising outcomes on mixed contaminant removal were reported by Reddy et al. [22]. Scientific investigations have provided insights suggesting that the chemical content and porosity of biochar contribute to its capability to interact with and thus trap P. For instance, the 73% of P removal efficiency by the biochar studied by Yao et al. [26] was attributed to its content of periclase (MgO). Various studies reported through investigations with Fourier-transform infrared (FTIR) spectroscopy, X-ray diffraction (XRD), and solid-state 13C nuclear magnetic resonance (NMR), the development of aromatic functionality and porosity in biochar produced under pyrolysis conditions of >400 °C. Therefore, it is possible that, if added to sand or soil filters, it could boost their performances, which we previously tested in a field pilot-scale treatment system [27]. In order to control and measure the treatment performance of certain parameters we also arranged a laboratory column experiment with real wastewater. The aim of this study was, therefore, to test the effectiveness of adding biochar to sand and gas–concrete filters on turbidity, phosphorus, and dissolved organic carbon (DOC) removal from wastewater.

## 2. Materials and Methods

### 2.1. Source of Filter Materials and Wastewater

The main materials were sand (Sa), biochar (C), and gas concrete (GC). The Sa was a fluvial-type soil obtained from the company Hakungekrossen AB (Sweden). The GC was supplied by Ecofiltration Nordic AB (with brand name Sorbulite®). Detailed description of this material can be viewed in Renman and Renman [28]. Information from the C supplier (Skogens kol AB, Sweden) indicated that it was produced from birch, aspen, and alder wood chips, which underwent a pyrolysis condition of 500 °C through a wagon retort process. The wastewater was obtained from the outlet of a tank receiving and treating raw wastewater from four households in Garns Ösby, a village located 35 km away from Stockholm (coordinates: 59.569649, 18.267817).

### 2.2. Preparation and Pre-Experiment Analysis of Materials

Prior to the experiment, the filter materials were crushed and sieved to required particle sizes (2–4 mm). Samples of the materials were used to determine their pH, bulk density ($\rho_b$), and porosity (ø). For pH, the samples were taken from prepared 1:20 solid:deionized water mixtures [26,29]. After adding a known volume of water ($V_w$) to a known volume of sample ($V_s$) in a graduated cylinder and then subtracting the measured volume of the mixture from the theoretical volume ($V_w + V_s$) to obtain the pore volume ($V_p$), ø was estimated as the ratio of $V_p$ to $V_b$ (bulk volume). Though it was important to obtain the hydraulic retention time (HRT) under unsaturated conditions, it was not easy to measure that for GC and C as they absorbed and retained the water until they had been fully wetted. The HRT was estimated from the time it took for each wetted filter bed to start discharging after it was dosed.

### 2.3. Preparing and Operating the Packed Bed Reactors (PBR)

The materials were packed in similar 12 columns (made of polyvinyl chloride, 0.7 m height, 0.045 m inner diameter). A 0.08 m layer of pebbles was first filled into each column to support the materials. A porous kitchen scouring pad was placed on top of the pebbles to prevent fine particles from passing through and clogging the outlet. The first three columns were filled with separate media, that is, Sa, GC, and C (each to 0.50 m depth) to serve as state-of-the art reference filters. The next set of three columns was filled with Sa (0.30 m) topped with C (0.20 m) while columns 7–9 were each filled with GC (0.30 m) topped with C (0.20 m) to serve as C-fortified Sa (SaC) and GC (GCC) filters. In order to obtain how both the filters would behave if all present, equal amounts of GC (0.1 m) and C (0.1 m) were added to Sa of 0.3 m depth to prepare SaGCC (sand-gas concrete-biochar) reactors. These were mainly tested on P removal only. Another scouring pad was placed on top of the media to facilitate distribution and pretreatment of the influent wastewater (Figure 1).

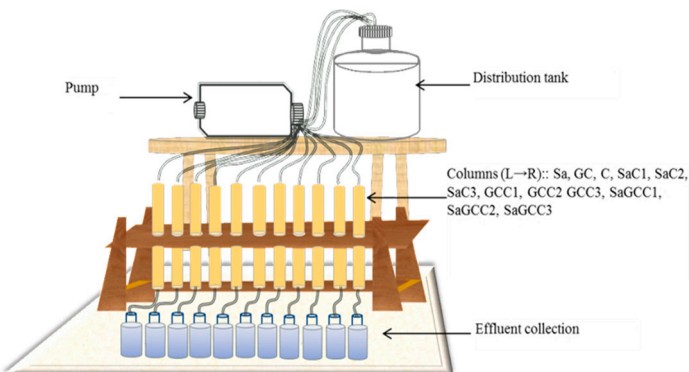

**Figure 1.** Packed bed reactor design and operation (Sa, sand; GC, gas concrete; C, biochar; SaC, sand + biochar; GCC, gas concrete + biochar; SaGCC, sand + gas concrete + biochar).

### 2.4. Operating the Columns

A timer-regulated pump was used to draw pretreated wastewater from a container placed on an elevated platform to distribute it by gravity onto the filters in timed intervals. From recommendations by the Swedish Environmental Protection Agency SEPA [30] (that is to use a hydraulic loading rate, HLR, of 40–50 $L/m^2/d$ for primary treatment and up to 100 $L/m^2/d$ for effluent polishing), 0.063 L/d HLR, that is, an equivalent of 40 $L/m^2/d$ or specific velocity of 4 cm/d, was adopted. This loading regime was maintained during the laboratory experiment of 399 days.

### 2.5. Sampling and Performance Analysis

Sampling of both the influent and effluents was made at least once per week. The samples were analyzed for pH, turbidity, orthophosphate ($PO_4^{3-}$), total phosphorus ($P_{tot}$), and dissolved organic carbon (DOC). Using the SensION$^{TM}$ PH31 (Hach$^®$) and 2100P ISO Turbidity meter (Hach$^®$), measurements of pH and turbidity, respectively, were made just after the sampling. The samples were frozen for later analyses of $PO_4^{3-}$ and $P_{tot}$ with the AutoAnalyzer 3 instrument (Seal Analytical Ltd.) and DOC (Shimadzu TOC-L Series, Shimadzu Corporation). Prior to $P_{tot}$ and DOC analysis, the samples were pooled together to make representative monthly samples into acid-washed plastic bottles. In total, 13 samples of the inlet and effluents were obtained. The $P_{tot}$ analysis followed a day after a two-hour autoclave treatment of the mixture of samples with $H_2SO_4$ (aq) and $K_2S_2O_8$ (aq). The results obtained from triplicate columns were averaged. A mass balance method was used to estimate the amount sorbed and the removal efficiencies (*E%*) of each filter bed. *E* was calculated using the relation:

$$E = \left(1 - \frac{C_o}{C_i}\right) * 100 \tag{1}$$

where $C_i$ and $C_o$ were the inlet and outlet concentration, respectively.

### 2.6. Statistical Analysis

For reliability of inferences to be made from the data, the data were first tested for normality by using the Shapiro–Wilk test tool of SPSS Statistics 21 (IBM). This was inferred from skewness and normality plots of the data. Since pH could help by indicating whether chemical reactions occurred in the treatment process or not, it was correlated to the efficiency of removal of P by each filter by use of a Pearson Product Moment correlation and regression analyses tools in SPSS. As the scope of the study, the significance of improvement in P and DOC removal by Sa after fortification with C was also evaluated. This was obtained by comparing means of the reference sand (Sa) and fortified sand (SaC) filters through independent t-tests. The probability (*α*-value) of the difference in the observed performance means occurring by chance was set at 5%.

## 3. Results and Discussion

### 3.1. Filter Characteristics

The result on observed filter properties is summarized in Table 1.

In preparing the media for use, the objective was for each of them to have ≥60% of the particles in the size range 2.0–4.0 mm so that they are not too fine or too coarse and thus less likely to let the water pass without treatment or clog within a short time. However, as shown in Table 1, more of the 0.1–2.0 mm fraction (and less of the 2.0–4.0 mm fraction) was found in C and GC than in Sa. This may have been the reason why they differed in hydraulic properties. It could actually be seen that some of the GC and C particles were so brittle that they tended to crumble as they were shaken to sieve them. This, as well as attrition between them, caused the percentage of the finer particles and pores in the media to increase [31]. Ultimately, the GC turned out to be the most (52%) while Sa was the least (35%) porous. Further evidence confirming this was the tendency of particles at the surfaces of those media to remain sodden longer after being dosed with the wastewater. It is suggested that

both inter- and intra-granular pores were present in those media. In an XRD study by Narayanan and Ramamurthy [32], both macro- (50 nm–50 μm) and micro- (<50 nm) pores were observed in the lattice of GC, a clear reason why it was so porous.

**Table 1.** Specifications of the filter media.

| Media | pH | $\rho_b$ (kg/m$^3$) | $d$ (mm) | ø | HRT (min) |
|---|---|---|---|---|---|
| Sa | 7.4 | 1.70 | 0.1–2.0 (39.8%) 2.0–4.0 (60.2%) | 0.35 | 90–78 |
| GC | 9.1 | 0.66 | 0.1–2.0 (43.4%) 2.0–4.0 (56.6%) | 0.52 | 127–99 |
| C | 8.0 | 0.54 | 0.1–2.0 (46.2%) 2.0–4.0 (53.8%) | 0.47 | 270–108 |
| SaC | 7.8 | n.d | 0.1–4.0 | 0.43 | 105–96 |
| GCC | 8.9 | n.d | 0.1–4.0 | 0.50 | 123–91 |

($\rho_b$, bulk density; d, particle diameter; ø, porosity; HRT, hydraulic retention time)

Related to porosity was HRT, which was observed to be shorter (<1–1.5 h) in the Sa than in the GC and C reactors (>1.5 h). It could be seen from the smoothness of particles of the Sa that it had few pores and thus offered less resistance to the percolating wastewater, hence its low HRT. However, as also confirmed by Jindo et al. [33], the sand with biochar (SaC) seemed to take up more water and retained it longer before discharging. Clearly, the biochar enhanced the structure and porosity of the Sa. The porosity (0.50) and HRT (2.0–1.5 h) of GCC were also found to be substantial. Many studies have associated the porosity of biochar to the manner in which it has been produced. For instance, Brewer et al. [34], Kim et al. [35], and Yargicoglu et al. [36] found out that biochar produced under pyrolysis condition of >400 °C is usually sufficiently porous for soil texture amendment. Under such conditions, volatiles and primary elements such as hydrogen, nitrogen, and sulphur, were driven out of its feedstock and thus left pores behind [36]. Though the BET (Brunauer-Emmett-Teller) surface area analysis of the biochar used in this study was not performed, observations made provided the clue that it was porous and thus could be used for wastewater filtration. By the end of the experiment, no imminent signs of clogging were observed despite the incidences of high turbidity. This proved that the adopted HLR (40 L/m$^2$/d) was appropriate for the treatment.

*3.2. Monitored Wastewater Qualities*

After 57 weeks of testing, 46 samples of the inlet and effluents had been obtained. Confidence intervals (and means with standard deviations in brackets) of the monitored parameters are presented in Table 2.

**Table 2.** Confidence and mean values (with standard deviation) of observed parameters.

| | Influent | Sa | C | GC | SaC | GCC |
|---|---|---|---|---|---|---|
| pH | 7.1–7.7 (7.5 ± 0.2) | 7.6–8.2 (7.7 ± 0.2) | 7.8–8.7 (8.0 ± 0.4) | 8.9–9.2 (9.1 ± 0.2) | 7.7–8.3 (7.9 ± 0.3) | 8.8–9.1 (8.9 ± 0.2) |
| Turbidity (NTU) | 29–428 (150 ± 210) | 1–3 (2 ± 1) | 1–2 (1 ± 0) | 0–1 (1 ± 0) | 1–2 (1 ± 0) | 0–1 (1 ± 0) |
| PO$_4^{3-}$ (mg/L) | 5.6–6.9 (6.1 ± 0.8) | 3.6–4.3 (3.9 ± 1.2) | 3.1–3.8 (3.5 ± 1.2) | 0.1–0.3 (0.2 ± 0.1) | 2.9–3.6 (3.3 ± 1.1) | 0.2–0.3 (0.2 ± 0.1) |
| P$_{tot}$ (mg/L) | 8.9–10.6 (10.0 ± 1.8) | 6.4–7.4 (7.6. ± 1.0) | 4.5–5.5 (6.4 ± 1.5) | 0.0–0.6 (0.3 ± 0.5) | 10.1–11.0 (6.2 ± 1.0) | 0.1–0.2 (0.1 ± 0.1) |
| DOC (mg/L) | 21.0–32.0 (25.3 ± 2.8) | 10.64–13.46 (12.1 ± 2.3) | 9.12–11.07 (10.1 ± 1.6) | 8.8–11.6 (9.4 ± 2.1) | 9.60–10.78 (10.2 ± 1.0) | 9.3–11.1 (10.0 ± 1.5) |

### 3.2.1. Turbidity and pH

It is known that grinding mineral-based matter can give rise to electrically unbalanced charges on their surface functional groups, for example, oxides ($O^{2-}$) or hydroxides ($OH^-$). As such, if such materials are immersed in solutions, they can bind some of the ions in the solution in order to become electrically balanced. This interaction has been found to be pH-dependent [37,38]. Therefore, as an important indicator of chemical reactions, the pH was monitored for comparisons of the behaviors of the reference and fortified filters in interacting with the wastewater constituents. Figure 2 shows how the pH changed with time.

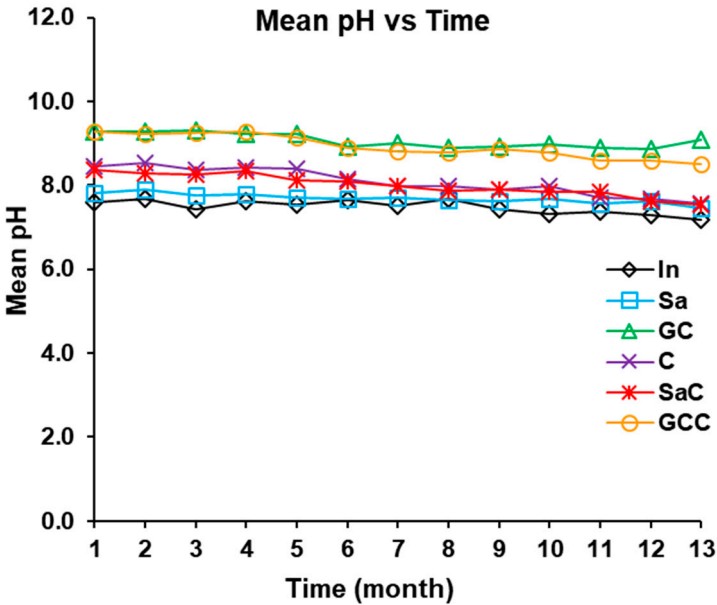

**Figure 2.** Mean pH change with time for the entire study duration (In, inlet; Sa, sand; GC, gas concrete; C, biochar; SaC, sand + biochar; GCC, as concrete + biochar).

As shown, fluctuations occurred in the pH of both the influent and effluents while a decreasing trend in pH of all effluents ensued from month 7. The mean influent pH (7.5) fell within an interval of pH 7.1–7.7. However, none of the lower bounds of intervals of all effluent pH measures fell within the influent pH range. This suggested that all the filters were still chemically active by the time the experiment was stopped. Further, the pH of effluents from all columns with GC was mostly >pH 8, a clear indication of the effect of high content of calcium in that material [39]. Some peaks occurred in influent pH whenever new wastewater was added to the ditribution tank thus showing that something in it spiked pH of the old water. This was probably due to the presence of alkaline substances such as detergents and dishwashing soaps. However, the pH in the storage tank tended to decrease slowly when it was kept longer. A most possible cause of this was the production of organic acids from the degradation of organic matter in the distribution tank [31]. Plots of turbidity as it changed with time are presented in log-scale in Figure 3.

The Figure 3 shows that the influent turbidity was unsteady throughout the entire study. However, it seemed that the fluctuations did not affect the turbidity in the effluents. The inlet turbidity range was 29–907 NTU (Nephelometric Turbidity Units) while that for all effluents was 1–6 NTU. All the filters maintained >95% efficiency in reducing it, even when the inlet turbidity was extremely high. This means that concentrations of suspended solids and dissolved matter (organic matter and colored substances) were greatly reduced by the filters. Though the removal efficiency of Sa was slightly lower than those of other filters initially, it became better as the experiment was continued. The sieving effect of its pores probably improved as they were slowly cemented by the biofilm resulting from accumulating solids and organic matter [31]. Despite that, no signs of clogging were imminent by the

end of the experiment. In general, the turbidity removal by all filters was acceptable according to WHO [40] guidelines on drinking water quality.

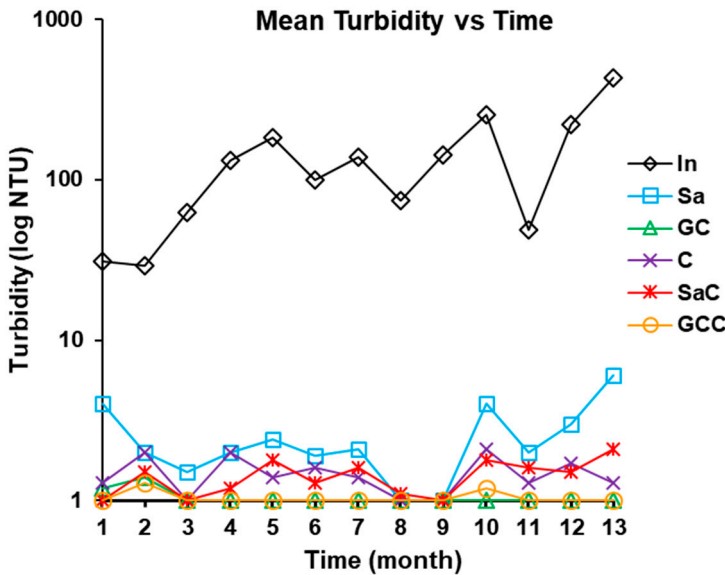

**Figure 3.** Mean influent and effluent turbidity trend during the entire study (In, inlet; Sa, sand; GC, gas concrete; C, biochar; SaC, sand + biochar; GCC, gas concrete + biochar).

### 3.2.2. Dissolved Organic Carbon (DOC) Removal

The DOC removal by septic systems is not often given attention, yet its impacts on the quality of surface waters are easily observable. As one of the nuisances in wastewater, it is important that the performance of septic systems in treating it is boosted. In addition to other parameters, the DOC was also monitored in this study. The variations of the loaded and effluent DOC concentrations are shown in Figure 4.

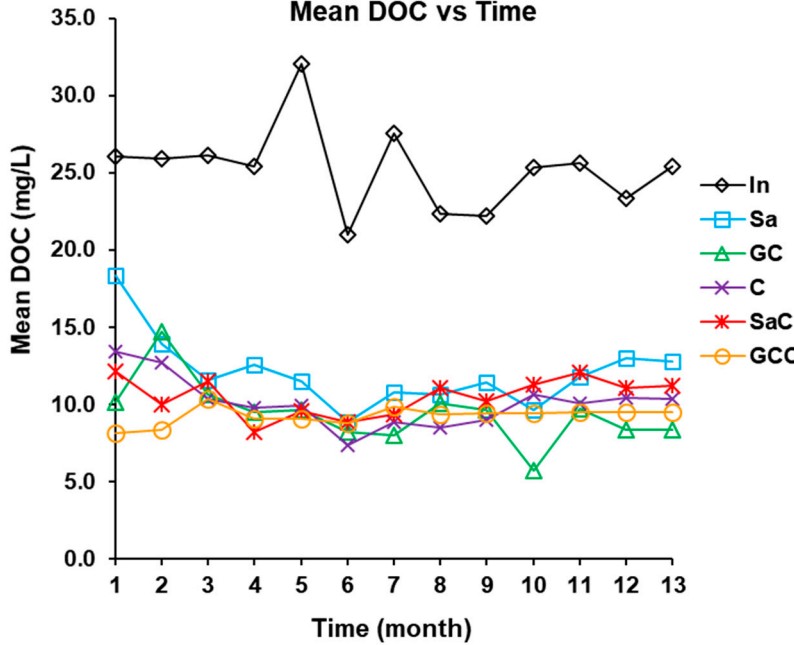

**Figure 4.** Dissolved organic carbon (DOC) variation during the entire study period (Sa, sand; GC, gas concrete; C, biochar; SaC, sand + biochar; GCC = gas concrete + biochar).

Unlike the other parameters, the influent DOC was almost stable during the first few months but started to fluctuate in later months. A peak was also observed in its trend during summer. This particularly happened around the time increased amounts of wastewater were generated from the households. It is with no doubt that the amount of organic waste flushed down the system also increased. The fluctuations and peaks of the DOC in the wastewater indicated that its concentration was affected by some factors [41]. For instance, the type of activities producing the wastewater, weather, aeration, etc., probably affected how much DOC reached and left the septic tank. The sampling time could also be related to the observed variations in DOC concentrations since this was not done on the same day of every week. In a study by Katsoyiannis and Samara [42], the DOC concentration in samples obtained on Mondays was found to be lower than that in samples obtained on other days. Another peculiar observation made in this current study was that after the wastewater in the distribution tank had been kept for longer time without topping it with new wastewater, its DOC concentration seemed to fall. Increased degradation of the DOC by microbial activity most possibly caused part of the influent DOC concentration to decline. In fact, the temperature range was 15.4–21.3 °C and 15.4–19.8 °C in the influent and effluents, respectively. It is clear from this that the surrounding temperature favored the microbial activities [43] during most of the time.

### 3.2.3. Filter Efficacy in DOC Removal

As presented in Table 2, an average of 25.25 mg/L was loaded onto each filter every month. On average, the reference Sa, C, and GC filters discharged about 12.1, 10.1, and 9.4 mg/L while the SaC and GCC released 10.6 and 10.0 mg/L. respectively. Therefore, the order of their performances could be written as Sa < C < SaC < GC < GCC. Moreover, when using monthly means, it was obtained from the statistics that the Sa, GC, C, SaC, and GCC were able to reduce the influent DOC by an average of 52%, 63%, 60%, 58%, and 61%, respectively. Basing on this data, it could be said that the Sa was the least while GC was most efficient at trapping the DOC. Even though the difference between the GC and GCC filter efficiency was almost insignificant, it was distinct between that of the reference Sa and fortified Sa filters. This suggested that the presence of the C boosted the capacity of the Sa in removing the DOC from the treated wastewater. Comparing this to findings in Katsoyiannis and Samara [42], who achieved 73% removal of DOC using activated sludge, it could be said that since the C had not been activated, the efficacies of the C and SaC filters were quite substantial.

As wastewater is continuously infiltrated through porous media, organic matter in it is accumulated on surfaces of particles of the media, forming a biomat. The biomat formed can contribute to reduction of DOC by absorption and biological degradation [44]. It is believed that the formation of biomat around the filter particles contributed to the substantial DOC removal efficiency of Sa, that is, >50%. Because of the high porosity of the GC and C media, it was highly likely that much of DOC was trapped in them by micro-pore filling mechanisms. In advanced treatment, whereby GAC is used [41], mechanisms of adsorption and ion exchange are normally responsible for removing dissolved substances. Even though the chemical composition of C was not analyzed in this study, its pH characteristics gave indications that it possessed some of these properties, hence the significant DOC removal by the Sa with C. However, its hydrophobicity may have played a part reducing chances of higher DOC removal in the beginning, that is, before it became colonized by fungi and thus fully wetted or pore-filled with water [45]. In general, the DOC removal by all filters with C was found to be substantial.

### 3.2.4. Phosphorus ($PO_4^{3-}$ and $P_{tot}$) Removal

Figures 5 and 6 shows how phosphorus (P) in its form as orthophosphate ($PO_4^{3-}$) and total phosphorus ($P_{tot}$) in the influent and effluents evolved with time. As Figure 5 shows, the influent $PO_4^{3-}$ concentration was almost constant during winter (December to April) but fluctuated during other times. A sharp drop in both the influent and effluents was observed towards the end of summer (September), a time during which the wastewater in the distribution tank was kept longer without adding fresh one to it. Part of the $PO_4^{3-}$ was probably bound to the settling solids while some was

taken up by micro-organisms [46]. Some $PO_4^{3-}$ peaks occurred in the influent and effluents of the Sa, C, and SaC reactors between June and August (summer) (Figure 5). An almost similar trend was found in the $P_{tot}$ trend (Figure 6). Those peaks occurred during the time when the number of inhabitants of the households generating the wastewater increased, showing that the amount of phosphorus-containing waste generated increased. During summer, approximately 3–4 people stayed in each of the households (4 in total). Thus, if we estimate using the typical per capita P loading of 2.0 g/day, then about 44 g of P was loaded into the system every day. Both Figures 5 and 6 also show that the $PO_4^{3-}$ and $P_{tot}$ in the effluents of Sa, C, and SaC steadily increased as the treatment process was continued. Their sorption sites (reactive species) were probably being exhausted.

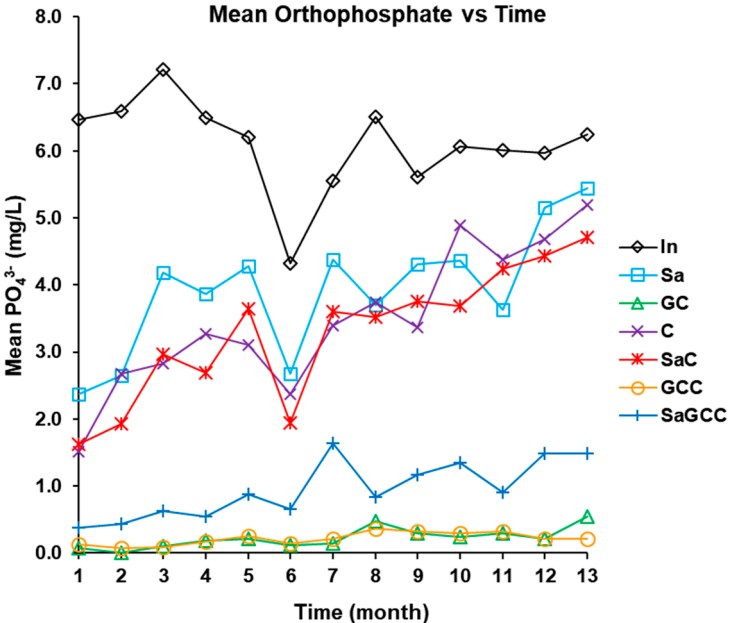

**Figure 5.** Phosphate trend during the entire study period (Sa, sand; GC, gas concrete; C, biochar; SaC, sand + biochar; GCC, gas concrete + biochar; SaGCC, sand + gas concrete + biochar).

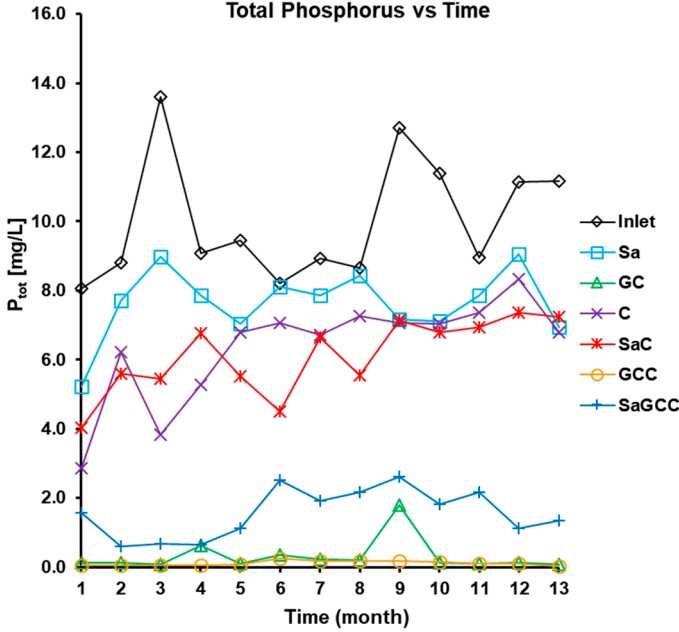

**Figure 6.** Total phosphorus trend during the entire study period (Sa, sand; GC, gas concrete; C, biochar; SaC, sand + biochar; GCC, gas concrete + biochar; SaGCC, sand + gas concrete + biochar).

In the case whereby all the media were combined in one column operated as a SaGCC reactor, Figure 5 shows that that combination was able to lower the influent P concentration to 1 mg/L. However, even though better than Sa, it seemed to be slightly less effective than the GC and GCC reactor. The average $PO_4^{3-}$ effluent concentration from that reactor was calculated to be 1 mg/L while its efficiency was about 86%, that is, 11% less effective than that of GC and GCC. Since C seemed to boost the Sa while its combination with GC was equally effective as the reference GC, then it could be said that the presence of Sa in SaGCC could be blamed for its lower efficiency.

## 4. Filter Efficacy at P Removal

The statistics showed that the mean influent $PO_4^{3-}$ concentration (6.1 mg/L) was within a 95% confidence interval (CI) of 5.6–6.9 mg/L while those for the reference reactors, that is, Sa (3.9 mg/L), C (3.5 mg/L), and GC (0.2 mg/L), were within 3.6–4.3 mg/L, 3.1–3.8 mg/L, and 0.1–0.3 mg/L, respectively (Table 2). On the other hand, the mean $PO_4^{3-}$ in the effluents of the fortified filters, that is, SaC (3.3 mg/L) and GCC (0.2 mg/L), were within CIs of 2.9–3.6 mg/L and 0.2–0.3 mg/L. As for $P_{tot}$, the influent and effluents of the reference Sa, C, and GC filters had mean concentrations of 10.0 mg/L and 7.6 mg/L, and 6.4 mg/L and 0.3 mg/L, respectively. Thus, it is clear from these findings that all reactors with GC were able to lower the influent P to <0.5 mg/L, thus satisfying the Swedish Environmental Protection Agency limit of <1 mg/L [7]. For the fortified filters, both Table 2 and Figures 5 and 6 show that while the GC filter seemed to be equally effective as its fortified counterpart, the SaC filter was a bit more effective than the reference Sa filter. Therefore, basing on this finding, it could be said that the C boosted the Sa in removing the $PO_4^{3-}$ from the wastewater.

In addition to statistical analysis of means and confidence intervals of the P in the influent and effluents, efficiencies in P reduction were also obtained (data not provided). By the end of the experiment, 25.14 L of wastewater had been treated through the different reactors. If we assume that each reactor received the same amount of $PO_4^{3-}$ (6.1 mg/L/month), then the total amount of $PO_4^{3-}$ loaded on each was about 152.7 mg. The analysis showed that Sa, GC, C, SaC, and GCC sorbed about 52.6, 147.1, 64.6, 69.9 and 147.2 mg of that amount, respectively. These corresponded to efficiencies of 35%, 96%, 42%, 45%, and 96%, respectively. This confirmed the conclusion that the GC filters were outstanding in P removal compared to those reactors with Sa and C.

Several factors are believed to have contributed to the observed variations in efficiencies of the different filters. For instance, the varying particle distributions led to differences in porosity, HRT, and specific surface areas. Renman and Renman [28] observed that a 2–4-mm particle size was sufficient for GC to achieve up to 100% P removal efficiency within just 1 day. The GC they used achieved about 93–99% P removal from wastewater. Findings in this current study (96% $PO_4^{3-}$ removal) seemed to agree with theirs. Morales et al. [47] also found out that for biochar to achieve up to 100% efficiency, it should be graded to particle size of ≤2 mm and also allowed to be in contact with wastewater for 72 h. As the biochar in this study had a lot of course particles and shorter HRT, it was possible for this to have an effect on its performance. In addition to its negative charge, which would possibly repel inbound negatively charged substances such as $PO_4^{3-}$, the hydrophobicity of the biochar possibly slowed part of the influent in penetrating into its particles. Further, since no insulation was provided, the low temperature condition in winter probably slowed down both chemical and biological processes by which the P was removed from the wastewater [43].

From the comparison of the reference and fortified filters, it was observed that while the GC was equally effective (96%) as its fortified pair (GCC), the SaC filter seemed to be more effective (43%) than the reference Sa filter (35%) in sorbing P. However, as was mentioned earlier, the efficiency of the reactors with Sa and C tended to reduce towards the end (Figures 5 and 6). This could be attributed to the effect of reduced effective filtration depth as well as limited intra-particular diffusion. On the other hand, the constantly high efficiency of the GC filters showed that there were still plenty of adsorbate-free sites and pores below the P mass transfer zone in them. As for SaC, the sites in the 0.2 C layer boosting the Sa were probably spent within a short time, thus letting much of the $PO_4^{3-}$

pass to the less-effective Sa layer below it and consequently reaching the bottom before much of it could be trapped. It is also possible that some of the chemical sites and pores were sealed by the established biomat.

## 5. Significance of Fortification with Biochar on DOC and P Removal by Sand

Further analyses were performed to ascertain if the observed higher efficacy of the fortified Sa filter (SaC) did not occur by chance. Firstly, the Levene's test of equality of variance gave a p-value of >0.05 for DOC and P removal by the Sa and SaC filters, thus showing that the variance in DOC and P concentrations in their effluents could not be assumed to be equal. Therefore, the significance level corresponding to an assumption of equal variances was used instead. Under this statistic the p-value for "t-test for equality of means" was 0.034 for DOC and 0.022 for P removal. These showed that there was a statistically significant difference between the mean DOC and also between the mean P in the Sa and SaC effluents. Also, the linear regression (LR) coefficients ($r^2$) between DOC and pH were 0.33, 0.66, and 0.70 for Sa, C, and SaC, respectively. The P removal efficiency-pH correlations (r) were 0.57, 0.82, and 0.85 for Sa, C, and SaC, respectively. This means that more than 65% of the change in DOC concentration of both C and SaC effluent could be explained by pH, whereas <50% did in the Sa effluent. Further, the LR model also revealed that 81% and 85% of variation in $PO_4^{3-}$ sorption in the C and SaC effluent, respectively, was explained by the variation in their pH measure while only 53% did for Sa. Except for turbidity, P-removal in an earlier field-scale study [27] related to this current study was a bit lower. The warmer room conditions (17–25 °C), less-mixed influent wastewater, small loaded wastewater volumes (0.021 L) per dose, etc., and smaller particle sizes (2–4 mm) used in the current study are believed to have contributed largely to the higher efficacies of the media. Nevertheless, the same was observed in both studies, that the fortification of sand filters with biochar significantly improved its performance in removing P and DOC.

## 6. Conclusions

This study successfully tested the idea of fortification of sand filters with biochar on boosting its performance in treating septic tank effluents (STE). Gas concrete (used alone or fortified with biochar) was also tested as an alternative to sand only. On turbidity, all the filters proved to be very effective, removing >95% of it from the STE. As for DOC, the efficiencies of the filters fell within the range 51–61%, thus relatively seeming to be equally effective. However, the SaC seemed to be 6% more effective (*p* = 0.03) than its reference pair while the GC-GCC pair did not differ in performance. Lastly, the GC proved to be the most capable at reducing the influent $PO_4^{3-}$ (6.1 mg/L) (by >95%) and thus keeping it below the SEPA limit of 1 mg/L. On the issue of fortification, there seemed to be significant improvement (*p* = 0.022) in the sand. Therefore, it was concluded that sand fortification with biochar could be a possible measure to improve the removal of turbidity and reduction of nutrients from STEs. However, further tests of the system of factors, such as varying hydraulic loading rate, order in which the materials were packed in the columns, particle size, etc., are recommended to find possible optimal designs of the filters. With further improvements of sand filter fortification, it could prove to be a possible method of upgrading septic systems to high environmental standards.

**Author Contributions:** Conceptualization, G.R. and E.K.; methodology, E.K., A.R., and G.R.; software, E.K.; validation and formal analysis, E.K.; investigation, E.K.; resources, E.K.; data curation, E.K.; writing, original draft preparation, E.K.; writing, review and editing, E.K. and A.R.; supervision, G.R. and A.R. All authors have read and agreed to the published version of the manuscript.

**Funding:** Botswana International University of Science and Technology (BIUST) Management funded this research by a PhD fellowship at KTH Royal Institute of Technology to E.K.

**Conflicts of Interest:** The authors declare no conflict of interest. The funder had no role in the design of the study; in the collection, analyses, or interpretation of data; in the writing of the manuscript, or in the decision to publish the results.

## Nomenclature

| | |
|---|---|
| PBR | Packed bed reactor |
| RF | Reactive filter |
| Sa | Sand |
| C | Biochar |
| GC | Gas concrete |
| GCC | Gas concrete + biochar |
| SaGCC | Sand + gas concrete + biochar |
| STU | Soil treatment units |
| STE | Septic tank effluent |
| DOC | Dissolved organic carbon |
| P | Phosphorus |
| $PO_4$ | Soluble reactive phosphorus |
| $PO_4^{3-}$ | Orthophosphate |
| $P_{tot}$ | Total phosphorus |
| Mg | Magnesium |
| N | Nitrogen |
| Ca | Calcium |
| Al | Aluminum |
| Fe | Iron |
| $O^{2-}$ | Oxides |
| $OH^-$ | Hydroxides |
| MgO | Periclase |
| $Ca_5(PO_4)_3OH$ | Calcium hydroxyapatite |
| $Ca(OH)_2$ | Slaked lime |
| SEPA | Swedish Environmental Protection Agency |
| HLR | Hydraulic loading rate |
| HRT | Hydraulic retention time |
| GAC | Granulated activated carbon |
| WHO | World Health Organization |
| OWT | On-site wastewater treatment |
| PTP | Package treatment plant |
| FTIR | Fourier-transform infrared |
| XRD | X-ray diffraction |
| NMR | Nuclear magnetic resonance |
| CI | Confidence interval |

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
