# Peer review of "Filter Media-Packed Bed Reactor Fortification with Biochar to Enhance Wastewater Quality"

_applsci, doi:10.3390/app10030790_

Round 1

Reviewer 1 Report

Kholoma et al. report on the effect of biochar on wastewater cleaning efficiency of various filter media-packed bed reactors.

The manuscript is well-written and the results are clearly presented. There are only a few comments which in my opinion could improve this version of the manuscript.

The most important issue is the lack of thorough characterization of the biochar used. The authors indicate through various published papers that the structure of biochar is drastically affected by the production process meaning that different phosphate capture aptitudes should be expected for differently produced biochars.

Given this, I believe the authors should provide a more careful characterization of the utilized material. Questions as the following should be answered

-is their biochar charged?

-are there heteroatoms included? If yes to which extent? Could this infuence the trapping efficiency of biochar towards phosphate anions?

-what is the adsorption efficiency of pure biochar towards phosphate?

-what are the main interactions of biochar and phosphates or organic matter anticipated?

I believe that the manuscript will be more complete after the above points are addressed.

Author Response

See attached document where responses to both reviewer´s comments are found.

Reviewer 2 Report

I revised the paper titled “Filter media-Packed Bed Reactor Fortification with Biochar to enhance Wastewater Quality”. The authors explained the application of sand, biochar and gas concrete in filter (and their combination), in order to remove pollutants from wastewater. The structure of the paper is simple to read and the scientific language is correct. The content is appropriate for publication on your Journal and the English is good. I suggest minor revisions. My minor comments are the following:

Introduction - from line 40 to line 53. A small number of references is provided in this section despite the high number of assertions. For instance, you said that the drawback of GAC is that it is very costly…. I suggest you a recent publication if it will be useful for you https://doi.org/10.1016/j.psep.2019.10.022 where activated carbon costs are monitored. I suggest repeating the meaning of the abbreviations GC- GCC- SaC in the caption of all figures to make them clearer. Figures: all ordinate axes should start from 0. The numbers of figures are not correct. There are two Figures 1… Please, correct this aspect. Please, provide a more in-depth description of what may be the reasons for different results using the different combinations of materials. There is a very high number of abbreviations. Please, provide a nomenclature at the end of the manuscript.

Author Response

See attached document where response to the comments is found

Round 2

Reviewer 1 Report

The authors have addressed all issues pointed out during the first revisions round and they have provided responses to all questions. I therefore believe it can be accepted in its current form.